# Shared and distinct roles of Esc2 and Mms21 in suppressing genome rearrangements and regulating intracellular sumoylation

**Raymond T. Suhandynata**[1], **Yong-Qi Gao**[1], **Ann L. Zhou**[1], **Yusheng Yang**[1], **Pang-Che Wang**[1], **Huilin Zhou**[1,2]*

**1** Department of Cellular and Molecular Medicine, University of California School of Medicine, San Diego, La Jolla, California, United States of America, **2** Moores-UCSD Cancer Center, University of California School of Medicine, San Diego, La Jolla, California, United States of America

\* huzhou@health.UCSD.edu

**Data Availability Statement:** All relevant data are within the paper and its Supporting information files.

## Abstract

Protein sumoylation, especially when catalyzed by the Mms21 SUMO E3 ligase, plays a major role in suppressing duplication-mediated gross chromosomal rearrangements (dGCRs). How Mms21 targets its substrates in the cell is insufficiently understood. Here, we demonstrate that Esc2, a protein with SUMO-like domains (SLDs), recruits the Ubc9 SUMO conjugating enzyme to specifically facilitate Mms21-dependent sumoylation and suppress dGCRs. The D430R mutation in Esc2 impairs its binding to Ubc9 and causes a synergistic growth defect and accumulation of dGCRs with mutations that delete the Siz1 and Siz2 E3 ligases. By contrast, *esc2-D430R* does not appreciably affect sensitivity to DNA damage or the dGCRs caused by the catalytically inactive *mms21-CH*. Moreover, proteome-wide analysis of intracellular sumoylation demonstrates that *esc2-D430R* specifically down-regulates sumoylation levels of Mms21-preferred targets, including the nucleolar proteins, components of the SMC complexes and the MCM complex that acts as the catalytic core of the replicative DNA helicase. These effects closely resemble those caused by *mms21-CH*, and are relatively unaffected by deleting Siz1 and Siz2. Thus, by recruiting Ubc9, Esc2 facilitates Mms21-dependent sumoylation to suppress the accumulation of dGCRs independent of Siz1 and Siz2.

## Introduction

Segmental duplications are "at-risk" DNA sequences that can cause genome rearrangements through non-allelic recombination pathways. Previous studies demonstrate that specific pathways are involved in preventing the accumulation of duplication-mediated genome rearrangements [1–3]. These pathways involve genes that act during DNA replication and repair, and genes that are involved in post-translational protein modifications. Among the latter, we found that modifications by **S**mall **U**biquitin-like **MO**difer (SUMO) play a highly specific and significant role in suppressing duplication mediated gross chromosomal rearrangements (dGCRs) [4, 5]. *Saccharomyces cerevisiae* expresses three mitotic SUMO E3 ligases, Siz1, Siz2

**Funding:** RTS was supported by a postdoc fellowship from NCI T32 CA009523. HZ is supported by NIH RO1 GM116897, S10 OD023498 and University of California Faculty seed grant. The funders had no role in study design, data collection and analysis, decision to publish, or preparation of the manuscript.

**Competing interests:** The authors have declared that no competing interests exist.

and Mms21 [6, 7]. Inactivating all three SUMO E3 ligases results in lethality [8], like the deletions of SUMO (*SMT*3), the sole E1 (*AOS1-UBA2*) and E2 *(UBC9)* enzymes [9], suggesting that these SUMO E3 ligases are necessary for controlling intracellular sumoylation. Although deletion of *SIZ1* and *SIZ2* results in a relatively modest increase in dGCRs, inactivating the Mms21 E3 ligase through point mutations in its catalytic domain (*mms21-CH)* leads to a highly specific accumulation of dGCRs [4, 5]. Moreover, combining *mms21-CH* with either *siz1Δ* or *siz2Δ* leads to a further increase in the rate of accumulating dGCRs, indicating the partially redundant roles of these SUMO E3 ligases [4]. This raised key questions that have yet to be understood. First, how do the SUMO E3 ligases, especially Mms21, target their substrates in the cell? Second, do mutations affecting how Mms21 targets its substrates cause genome instability? And if so, what are the potential substrates involved? To address these questions, a better understanding of how SUMO E3 ligases target their substrates in the cell is needed.

A quantitative SUMO proteomics technology was previously developed by our group to evaluate the relationships between the SUMO E3 ligases and their substrates [4]. These studies revealed that Siz1 and Siz2 control the bulk of intracellular sumoylation and their main targets are involved in gene transcription and related processes [4, 8, 10, 11]. By contrast, the Mms21 E3 ligase targets a smaller subset of proteins that primarily function in chromosomal maintenance [4, 7, 12]. Despite these disparate substrate preferences, there exists ample overlap between the substrates of these the three SUMO E3 ligases. For example, the Structure Maintenance of Chromosome (SMC) proteins were shown to be preferentially targeted by Mms21, and this preference was based on the finding that *mms21-CH* reduces their sumoylation levels by 2- to 4- fold [4]. Thus, Siz1 and Siz2 also contribute to the sumoylation levels of the SMC proteins, albeit to a lesser extent. This partial substrate preference of Mms21 extends to its other targets such as the Mini-Chromosome Maintenance (MCM) complex [12], which acts as the catalytic core of the DNA replicative helicase [13–15]. MCM sumoylation has been detected in both unperturbed cells and in response to DNA damage [16, 17]. Apart from Mcm6, which appears to be targeted by Siz1 and Siz2, Mms21 preferentially targets several MCM subunits in the unperturbed cell, including Mcm2, Mcm3 and others [12], suggesting that sumoylation of different subunits in the same complex could have different functions. Understanding how Mms21 targets its substrates may help to discern their distinct functions.

It is presently unknown how Mms21 targets its substrates in cells. Of interest here is Esc2, a protein with two SUMO-like domains (SLDs) [18]. Esc2 has been shown to specifically prevent the accumulation of dGCRs [1, 4]. Interestingly, double mutants combining *esc2Δ* and *mms21-11*, an allele that contains a pre-mature stop codon in Mms21 [7], accumulate dGCRs at a similar rate compared to *mms21-11* alone [4]. One caveat of this observation is that *mms21-11* may affect its function as an integral component of the Smc5-6 complex. Nevertheless, this finding raised the possibility that Esc2 might work in the same pathway as Mms21 in suppressing dGCRs. In support of this idea, *esc2Δ* was shown to cause global sumoylation changes that are similar to those caused by *mms21-CH* [4]. However, this earlier proteomic study did not identify sumoylation of DNA replication components, such as the MCM complex, due to their relatively low levels in unperturbed cells. Thus, whether Esc2 facilitates Mms21 to sumoylate DNA replication components has yet to be addressed.

Besides acting to suppress dGCRs in unperturbed cells, Esc2 has been shown to participate in DNA repair, sister chromatid cohesion, gene silencing and others [19–25]. The DNA repair function of Esc2 has been extensively studied using cells treated with DNA alkylating agents and these studies indicate that aberrant recombination intermediates accumulate in the *esc2Δ* mutant, similar to those seen in the *mms21* and *sgs1* mutants [20, 24, 26, 27]. More recently, Esc2 was shown to bind specific DNA repair structures and recruit DNA repair enzymes [23, 24], indicating that Esc2 may assemble these enzymes to facilitate DNA repair. The study of

separation-of-function *esc2* mutants is therefore needed to understand these diverse functions. Esc2 has two SLDs in its C-terminus, which are conserved among its fungal orthologs [18, 28, 29]. The C-terminal SLD2 domain of *S. pombe* Rad60, the ortholog of Esc2, has been shown to interact with the Ubc9 E2 enzyme [28, 29]. These studies demonstrated that a mutation affecting this interaction caused similar defects to those of *nse2-SA*, a mutation that inactivates the *S. pombe* ortholog of Mms21, suggesting that *S. pombe* Rad60 recruits Ubc9 to facilitate the function of Nse2. However, whether the *rad60* mutation specifically affects the sumoylation levels of Nse2 targets in the cell has not been determined. Interestingly, the interaction between *S. pombe* Rad60 and Ubc9 appears to be conserved in *S. cerevisiae*, considering that a yeast two-hybrid study has detected an interaction between Esc2 and Ubc9 [20]. Collectively, these studies suggest that Esc2/Rad60 recruits Ubc9 to specifically facilitate the functions of Mms21/ Nse2. Here, we combine genetic and proteomic analyses to explore this idea further, demonstrating that Esc2 recruits Ubc9 to specifically regulate how Mms21 targets its substrates, and that this Esc2-Ubc9 pathway acts in parallel to the other SUMO E3 ligases Siz1 and Siz2.

## Results

### Esc2 and Mms21 have distinct and shared roles

Our previous study used the *mms21-11* allele to evaluate the genetic relationship between Esc2 and Mms21 in suppressing GCRs [4]. To precisely determine the relationship between Esc2 and the SUMO ligase activity of Mms21, we examined the *mms21-CH* mutant, in which the conserved cysteine and histidine residues in its catalytic RING domain are substituted with alanine [5, 8]. Several lines of evidence suggest that Esc2 and the catalytic activity of Mms21 play both distinct and shared roles. First, spot assay shows that the *mms21-CH* mutant is more sensitive than *esc2Δ* to hydroxyurea (HU) (Fig 1A), which causes DNA replication stress by depleting dNTP levels in the cell. Moreover, the *esc2Δ mms21-CH* double mutant grows slower than either single mutant and is hypersensitive to 50 mM HU, while the *esc2Δ* mutant is not appreciably sensitive at this dosage of HU. This indicates that Mms21 plays a more important role in dealing with DNA replication stress compared to Esc2, and that they share a partially overlapping role in maintaining cell growth. Second, Esc2 and Mms21 appear to act in the same pathway to regulate intracellular sumoylation, and this function is best evaluated using cells lacking the Siz1 and Siz2 SUMO E3 ligases in which Mms21 is the sole remaining mitotic E3 ligase [9]. If Esc2 acts in the same pathway as Mms21 to regulate sumoylation, then *esc2Δ* is predicted to be lethal in the *siz1Δ siz2Δ* mutant. To test this, plasmid shuffling was performed, showing that *esc2Δ* is lethal in the *siz1Δ siz2Δ* mutant (Fig 1B), suggesting that Esc2 and Mms21 may act in the same pathway to regulate intracellular sumoylation. Third, previous studies have shown that mutations to Esc2 and Mms21 cause specific accumulations of dGCRs and suggested that they may act together [1, 4]. Combining *mms21-CH* and *esc2Δ* results in a 5- to 10- fold increase in the rate of dGCRs compared to that of either single mutant, respectively (Fig 1C). Thus, despite their similarities, there are differences in the manner in which Esc2 and Mms21 suppress dGCRs. To explore these differences further, we tested several genes that are required for the accumulation of dGCRs in the *mms21-CH* mutant [5], which include *POL32*, *RAD52*, *RAD9* and *RRM3*. Combining *pol32Δ* and *rad52Δ* with *esc2Δ* results in a similar reduction in the rate of dGCRs, which mirrors their effects in the *mms21-CH* mutant (Fig 1D). Thus, like the *mms21-CH* mutant, formation of dGCRs in the *esc2Δ* mutant occurs through Rad52- and Pol32- dependent break-induced replication [5]. Similarly, deletion of *RAD9* suppresses the accumulation of dGCRs in the *esc2Δ* mutant, which is again mirrored in the *mms21-CH*, confirming the role of the Rad9 DNA damage checkpoint in promoting the formation of dGCRs [5]. On the other hand, *rrm3Δ* results in a further increase

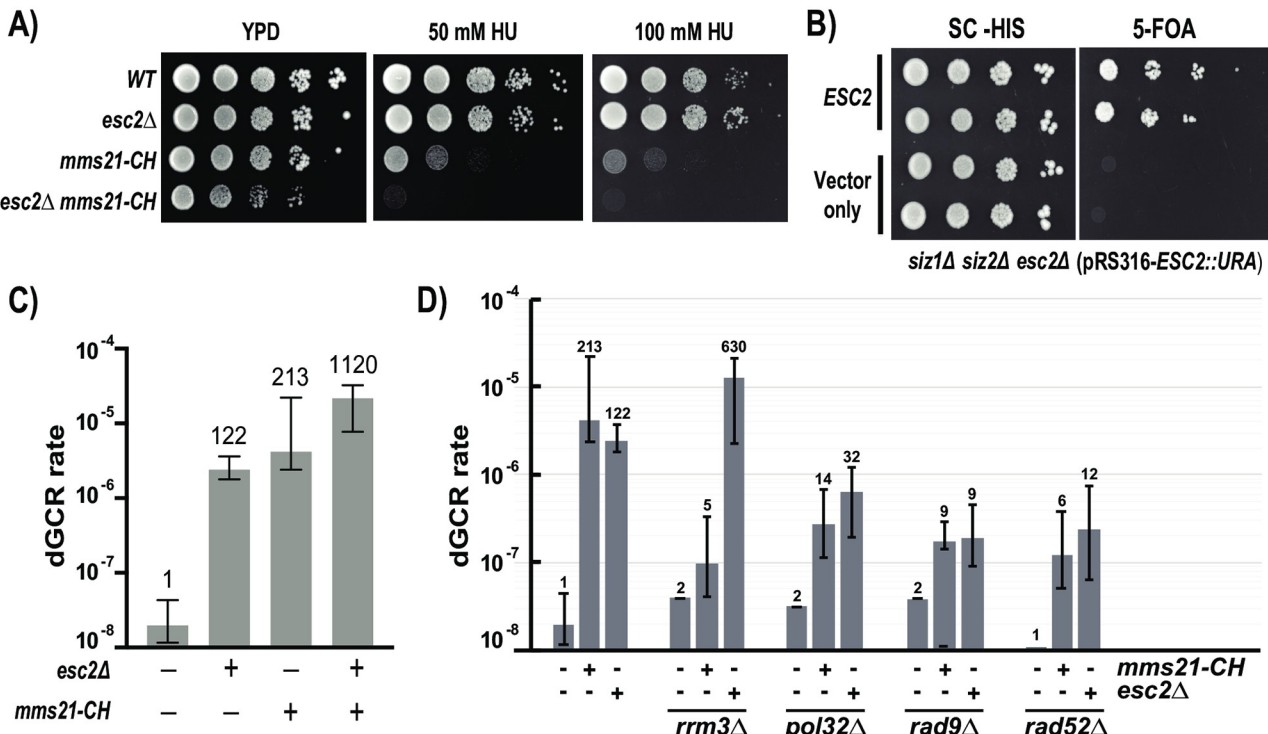

**Fig 1. Esc2 and Mms21 have distinct and shared roles in maintaining cell growth and suppressing dGCRs.** A) Spot assays to measure hydroxyurea (HU) sensitivity (50 mM and 100 mM) of WT, *esc2Δ*, *mms21-CH* and *mms21-CH esc2Δ* mutants. The YPD plate is shown to show cell growth in the absence of drug. B) 5-FOA plasmid shuffling of a URA3 plasmid expressing *ESC2* demonstrates the effect of evicting Esc2 in the *siz1Δ siz2Δ esc2Δ* triple mutant. C) dGCR rates of WT, *esc2Δ*, *mms21-CH* and *esc2Δ mms21-CH* mutants, measured via fluctuation analysis. D) dGCR fluctuation analysis demonstrating the effect of combining *esc2Δ* with *rad52Δ*, *pol32Δ*, *rrm3Δ* and *rad9Δ*. Error bars represent the upper and lower bounds of the 95% confidence interval with fold changes relative to WT shown above each respective analysis. For reference, *mms21-CH* specific data were taken from our previous studies.

in the rate of dGCRs in the *esc2Δ* mutant, while it suppresses the dGCRs of *mms21-CH* [5]. This suggests that Esc2 may have other functions that are influenced by Rrm3. Taken together, these results show that Esc2 and the Mms21 SUMO E3 ligase have both shared and distinct functions, and the analysis of separation-of-function *esc2* mutants is required to better understand them.

## A conserved di-phenylalanine motif in Esc2's N-terminus regulates its stability and function

To gain insights into Esc2, we performed sequence alignment of Esc2's fungal orthologs. This reveals a highly conserved di-phenylalanine motif in the unstructured N-terminus of Esc2 (Fig 2A). To test its role, both phenylalanine residues are substituted by alanine in the chromosomal *ESC2* locus. Additionally, a C-terminal TAF tag is fused to both the WT and Esc2-2FA proteins, allowing their protein levels to be determined. Western blot analysis shows that the level of the Esc2-2FA protein is approximately 3-fold lower than that of WT Esc2 (Fig 2B), indicating that this mutation partially reduces Esc2's expression level in the cell, despite the fact that it affects the unstructured region of Esc2. To evaluate the function of this conserved di-phenylalanine motif, a plasmid-shuffling experiment was performed, again using the *siz1Δ siz2Δ* strain background in which the function of Esc2 is essential for cell viability (Fig 1).

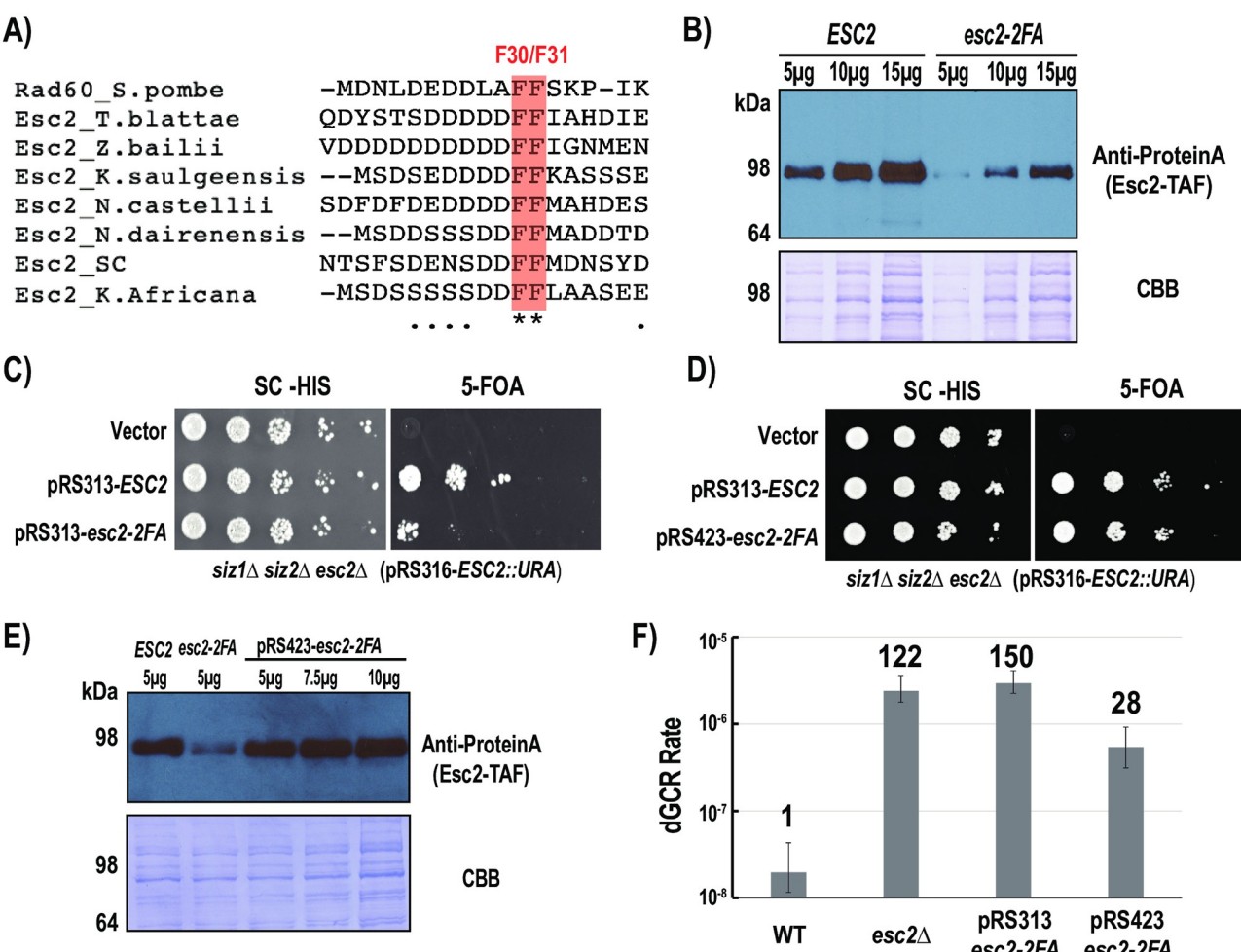

**Fig 2. A conserved di-Phe motif in the N-terminus of Esc2 is required for its stability and function.** A) Sequence alignment of fungal Esc2 ortholog highlighting the di-Phe motif across eight different fungi. B) 5-FOA plasmid shuffling of a URA3 plasmid expressing *ESC2* demonstrates the growth defect of Esc2-2FA when expressed from a centromeric plasmid (low-copy) in the *siz1Δ siz2Δ esc2Δ* strain background. C) Western blot demonstrating the expression level of the esc2-2FA mutant protein when expressed from a centromeric plasmid. D) 5-FOA plasmid shuffling demonstrates the effect of expressing Esc2-2FA from a high-copy 2-micron plasmid in the *siz1Δ siz2Δ esc2Δ* mutant background. E) Comparison of protein expression levels of endogenous WT, Esc2-2FA expressed from a centromeric plasmid and Esc2-2FA expressed from a high-copy plasmid. F) dGCR analysis of the *esc2-2FA* mutant and the effect of high-copy expression of Esc2-2FA in the *esc2Δ* mutant. Error bars represent the upper and lower bounds of the 95% confidence interval with fold changes relative to WT shown above each respective analysis.

Unlike *esc2Δ*, expressing Esc2-2FA is not lethal in the *siz1Δ siz2Δ* mutant, but it results in a dramatic cell growth defect (Fig 2C). This defect could be due to the lower expression level of Esc2-2FA (Fig 2B). To test this, Esc2-2FA is expressed from a high-copy 2-micron plasmid under its native promoter in the *siz1Δ siz2Δ* mutant, which significantly improves cell growth (Fig 2D). Moreover, this high-copy expression strategy appears to restore the expression level of Esc2-2FA to the WT level (Fig 2E), suggesting that reduction in the level of endogenous Esc2-2FA is responsible for the growth defect in the *siz1Δ siz2Δ* mutant background. Next, we evaluated the effect of *esc2-2FA* on the rate of accumulating dGCRs. Upon integration into its chromosomal locus, *esc2-2FA* causes a similar increase in the rate of accumulating dGCRs to the *esc2Δ* mutant (Fig 2F). Restoration of Esc2-2FA's protein level through the use of a high-copy plasmid partially prevents the accumulation of dGCRs in the *esc2Δ* mutant background,

suggesting that this conserved di-phenylalanine motif of Esc2 plays a partial role in suppressing dGCRs, besides stabilizing Esc2. Attempts to determine whether this di-phenylalanine motif of Esc2 mediates protein-protein interactions have so far yielded no additional insight. Thus, further study is needed to understand its role in partially suppressing dGCRs.

## Recruitment of Ubc9 by Esc2 acts parallel to that of Siz1 and Siz2

*S. pombe* Rad60, the ortholog of *S. cerevisiae* Esc2, has been shown to bind Ubc9 through its SLD2 domain [28, 29]. Yeast two-hybrid analysis showed that *S. cerevisiae* Esc2 binds to Ubc9 [20], suggesting that this interaction is conserved. Indeed, sequence alignment of the SLD2 domains of fungal Esc2 proteins reveals that E380 of *S. pombe* Rad60, which has been shown to mediate the Rad60-Ubc9 interaction, is conserved; and that the equivalent residue, D430 of *S. cerevisiae* Esc2, is an acidic residue like E380 of Rad60 (Fig 3A). To test its function, *esc2-D430R* was integrated into the chromosomal locus of *ESC2*, and it appeared that this mutation does not appreciably affect the protein levels of Esc2 in the cell (Fig 3B). To evaluate its effect on the presumed binding between Esc2 and Ubc9, we expressed an N-terminal Protein-A tagged Esc2, both WT and D430R, from bacteria, used IgG resin to capture these proteins and tested for Ubc9-binding. As shown in Fig 3C, *esc2-D430R* reduces the binding between Esc2 and Ubc9 to background levels, while the WT Esc2 protein robustly captures Ubc9. Thus, Esc2 binds to Ubc9 directly and the *D430R* mutation disrupts this interaction. Next, the effect of *esc2-D430R* on the growth of the *siz1Δ siz2Δ* mutant was tested, again using the plasmid-shuffling technique. Unlike the deletion of *ESC2*, *esc2-D430R* did not cause lethality, but it did cause drastically reduced cell growth (Fig 3D). To test this further, tetrad dissection was performed, showing that the *siz1Δ siz2Δ esc2-D430R* triple mutant spore grows significantly slower compared to the *esc2-D430R* single or *siz1Δ siz2Δ* double mutants (Fig 3E). Moreover, a spot assay confirmed that the *siz1Δ siz2Δ esc2-D430R* triple mutant is sensitive to HU, in addition to its impaired growth (Fig 3F). All together, these findings show that the conserved Esc2-Ubc9 interaction acts in a pathway parallel to that of Siz1 and Siz2, and that the *D430R* mutation specifically disrupts this interaction. These findings are similar to those observed for *S. pombe rad60-E380R* [28, 29], suggesting that the mechanism is conserved.

## Epistatic relationship between Esc2-Ubc9 and Mms21

These findings suggest that Esc2-Ubc9 and Mms21 have some shared functions. To evaluate this relationship, a spot assay was performed, demonstrating that the *esc2-D430R mms21-CH* and the *mms21-CH* mutants grow at a similar rate and are similarly sensitive to hydroxyurea (HU) (Fig 4A). We also examined the sensitivity of *esc2* and *mms21* mutants to methyl methanesulfonate (MMS). Although *esc2Δ* is sensitive to MMS, *esc2-D430R* is not (Fig 4B). Moreover, unlike *esc2Δ*, *esc2-D430R* does not appear to elevate the MMS sensitivity of *mms21-CH*, suggesting that the Esc2-Ubc9 interaction specifically acts in the same pathway as Mms21. The dGCR rate observed in *esc2-D430R* is moderately increased (~ 10 fold), compared to *mms21-CH* or *esc2Δ* [4], suggesting that recruitment of Ubc9 by Esc2 only partially contributes to Esc2's function in suppressing dGCRs. Similarly, combining *esc2-D430R* and *mms21-CH* does not appreciably affect the rate of accumulating dGCRs when compared to the *mms21-CH* mutant alone (Fig 4C). In contrast, a drastic increase in the rate of accumulating dGCRs is observed in the *esc2-D430R* when it is combined with the *siz1Δ siz2Δ* mutant. Thus, the Esc2-Ubc9 interaction specifically functions together with Mms21 to suppress dGCRs and acts in a pathway that is parallel to that of Siz1 and Siz2. Moreover, the overall stronger phenotypes of the *mms21-CH* mutant suggest that the Esc2-Ubc9 pathway plays a secondary role that assists Mms21 in performing its functions.

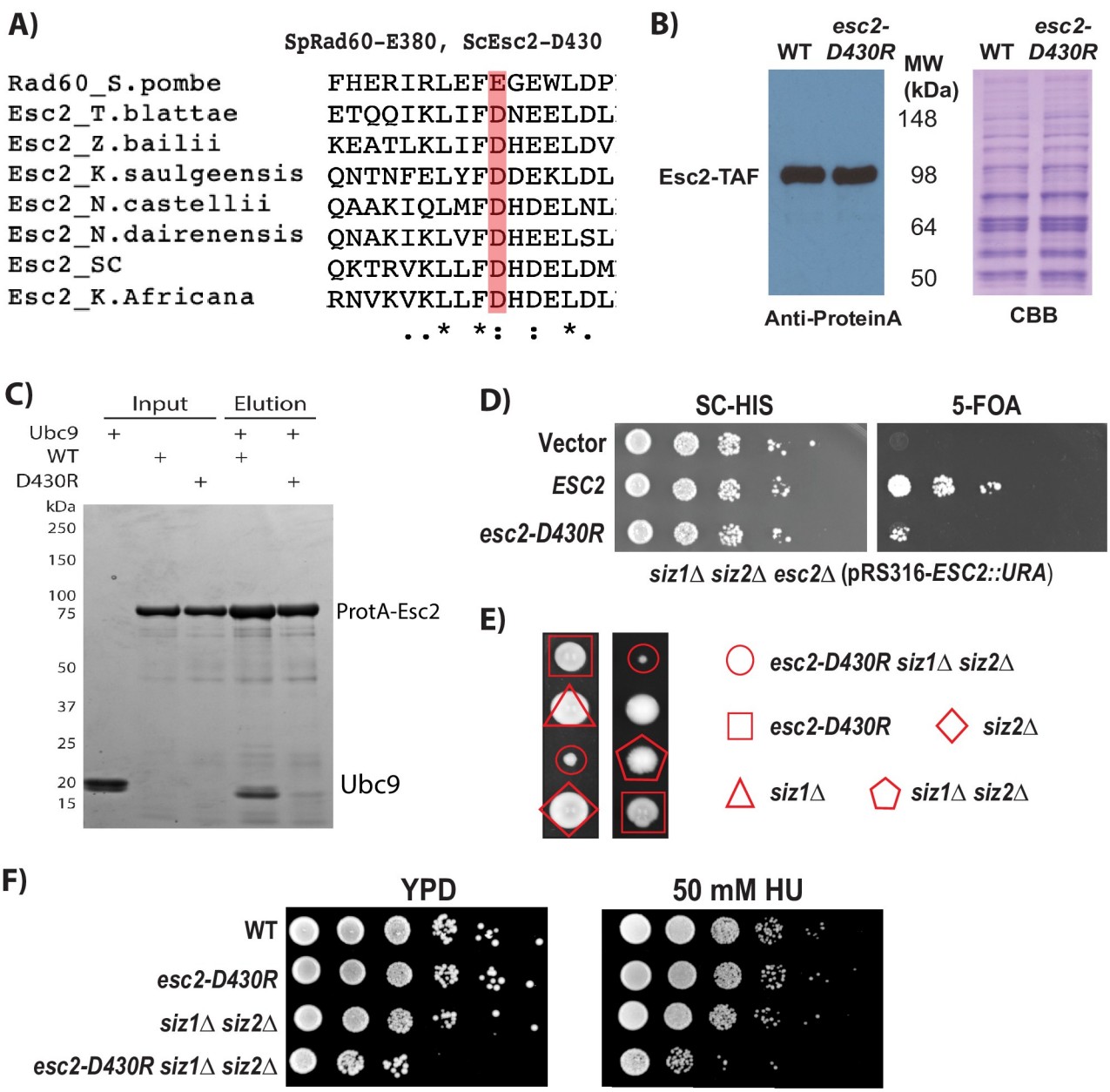

**Fig 3. Effect of *esc2-D430R* on cell growth and Ubc9 binding.** A) Sequence alignment of fungal Esc2 orthologs, highlighting the conserved D430 (ScEsc2) and E380 (SpRad60) residues. B) Western blot demonstrating protein expression levels of Esc2-D430R and WT Esc2. Endogenous C-terminal TAF tagged Esc2 proteins are detected by anti-Protein-A antibody, showing the D430R mutation does not appreciably affect the protein level of Esc2. C) Effect of the *esc2-D430R* mutation on Esc2 and Ubc9 binding. D) 5-FOA plasmid shuffling demonstrates the effect of *esc2-D430R* on the growth of the *siz1Δ siz2Δ esc2Δ* mutant, which is kept alive by a plasmid with WT *ESC2*. E) Representative tetrad dissection of diploids containing heterozygous mutations of *siz1Δ*, *siz2Δ* and *esc2-D430R*. F) Spot assays evaluating the growth properties of the *siz1Δ*, *siz2Δ* and *esc2-D430R* mutants.

## Role of Esc2-Ubc9 binding in regulating intracellular sumoylation

Deletion of Esc2 was previously shown to cause a wide range of perturbations to intracellular sumoylation, and these effects resembled those of the *mms21* mutant [4]. The most straightforward explanation is that Esc2 may recruit Ubc9 to facilitate Mms21-dependent sumoylation. To test this, we applied our previously described quantitative SUMO proteomic approach to

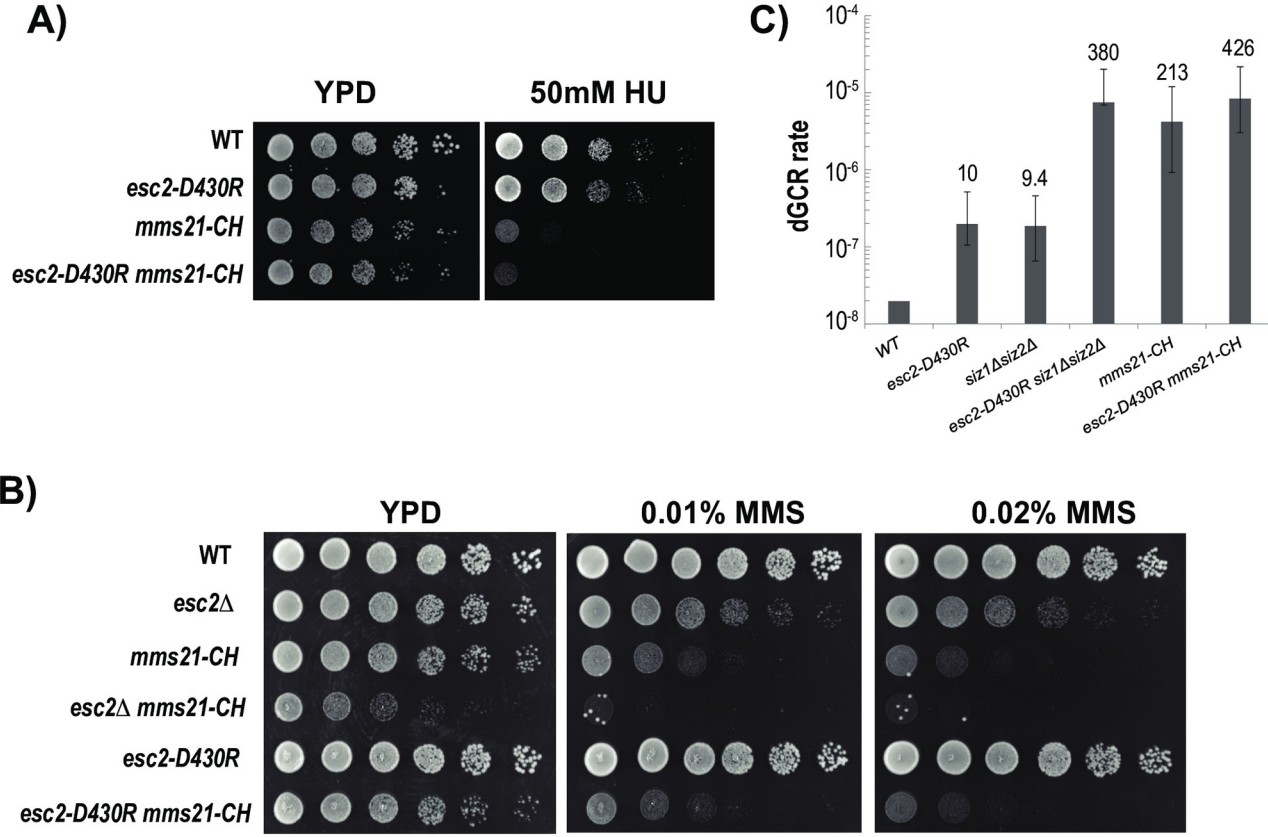

**Fig 4. Effect of *esc2-D430R* on DNA damage sensitivity and the rate of accumulating dGCRs.** A) Spot assay to evaluate the growth of the *esc2-D430R*, *mms21-CH* and *esc2-D430R mms21-CH* mutants in response to hydroxyurea (HU) treatment. B) Spot assay to evaluate the growth of *esc2* and *mms21-CH* mutants in response to methyl methanesulfonate (MMS) treatment. C) dGCR rates of *esc2-D430R* alone and together with *siz1Δ siz2Δ* and *mms21-CH* mutants. Error bars represent the upper and lower bounds of the 95% confidence interval with fold changes relative to WT shown above each respective analysis.

compare intracellular sumoylation levels in the WT and *esc2-D430R* mutant [4]. Compared to our previous studies [4, 12], more sumoylated targets were identified and quantified here (S2 Table), which could be due to the improved sensitivity of the MS instrument used. This allowed for a broader quantification of sumoylated proteins, where accurate quantification of each protein is based on the relative abundance of multiple unique peptides. Septins, a family of GTPases that play an important role in cytokinesis, are among the most abundant sumoylated proteins in yeast and have been found to be targeted primarily by Siz1 and Siz2 [30]. The *esc2-D430R* mutation leads to 2- to 4- fold increases in Septin sumoylation levels (Fig 5A), similar to the effect of *esc2Δ* [4]. Considering the redundant role of Esc2-Ubc9 and Siz1/Siz2 in maintaining cell growth (Fig 3), elevated sumoylation levels of the Septins could be attributed to a compensatory increase of Siz1/Siz2 activity in the *esc2-D430R* mutant, although other possibilities cannot be discounted.

The majority of sumoylated proteins reside in the nucleus of the cell. Starting from the nuclear membrane, multiple nuclear pore associated proteins have been found to be sumoylated and their sumoylation levels are not appreciably affected by *esc2-D430R* (Fig 5B). Like *esc2Δ*, *esc2-D430R* reduces the sumoylation levels of proteins in the nucleolus (Fig 5C), which include the RNA Pol-I subunits Rpa135, Roa190, as well as the nucleolar silencing complex

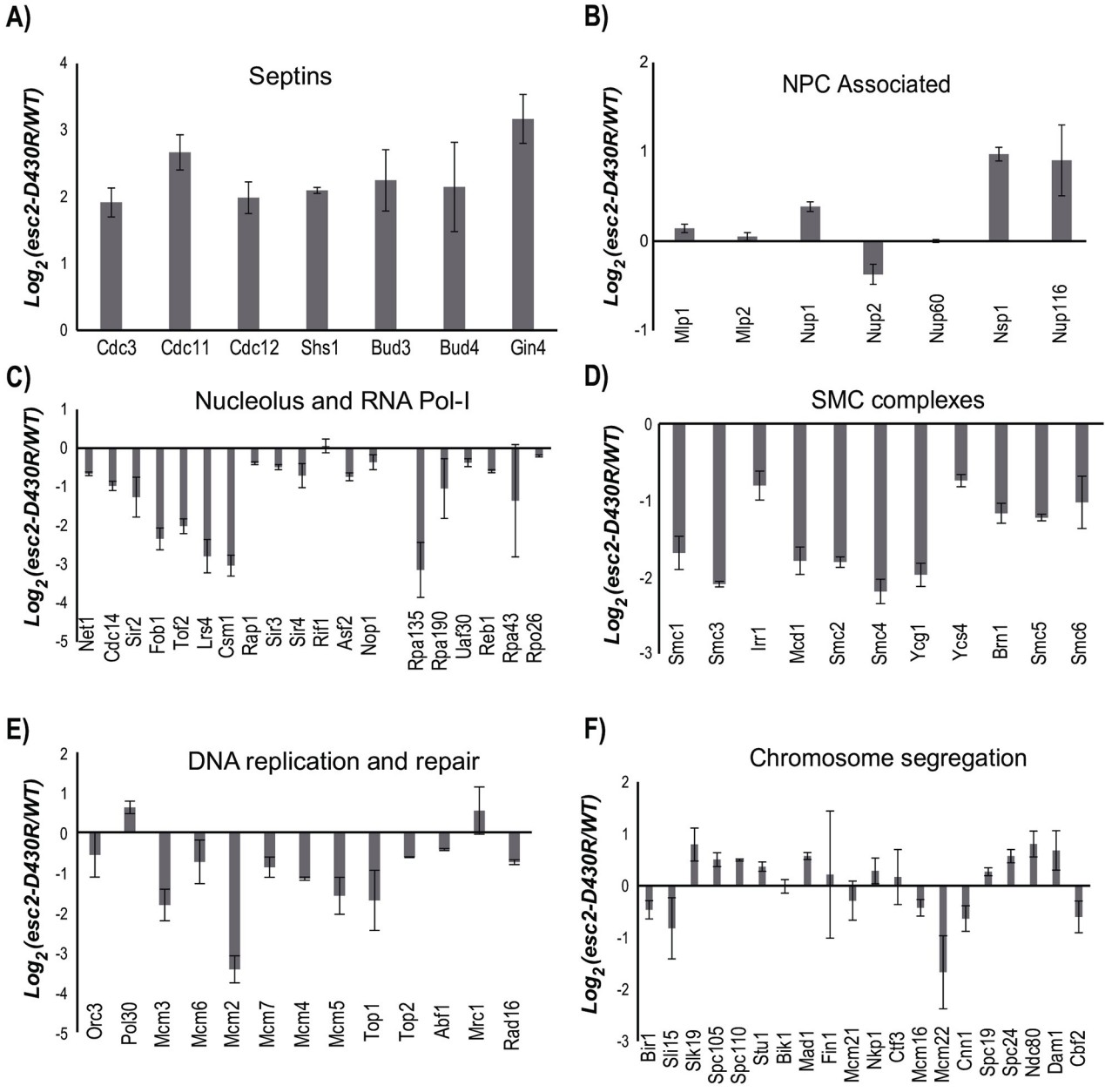

**Fig 5. SUMO proteomic approach to evaluate the effect of the *esc2-D430R* mutation on intracellular sumoylation via a comparison of the *esc2-D430R* mutant and WT cells.** Sumoylation levels of proteins in several functional groups are summarized here. A) Septins, B) NPC associated, C) Nucleolus and RNA Pol-I, D) the SMC complexes, E) DNA replication and repair and F) Chromosome segregation. Log₂ ratios of comparing the relative abundance of each protein in the WT vs the esc2-D430R mutant is shown, along with error bars representing the standard deviation calculated using multiple unique peptides of the same protein (see experimental method for details).

Lrs4-Csm1-Tof2, which has not been reported previously. Similarly, *esc2-D430R* leads to 2- to 4- fold reductions in the sumoylation levels of the SMC proteins and their associated subunits (Fig 5D). These effects resemble those observed for the *esc2Δ* and *mms21-11* mutants [4], suggesting that Esc2 recruits Ubc9 to facilitate Mms21-dependent sumoylation. Mms21 was shown to regulate sumoylation levels for all of the MCM subunits except for Mcm6 in unperturbed cells [12]. Whether Esc2 regulates MCM sumoylation through the recruitment of Ubc9

has not been determined. The improved sensitivity of our SUMO proteomic analysis enables the identification and quantification of the majority of MCM subunits, showing *esc2-D430R* causes 2- to 4- fold reductions in MCM sumoylation levels, except for Mcm6 (Fig 5E). These effects resemble those of *mms21-CH*, suggesting that Esc2 recruits Ubc9 to catalyze MCM sumoylation by Mms21. Additionally, we have identified and quantified sumoylation levels of many proteins involved in chromosome segregation for the first time, showing that *esc2-D430R* has a relatively modest effect on their sumoylation levels (Fig 5F). Finally, *esc2-D430R* does not strongly perturb the sumoylation levels of proteins involved in gene transcription and chromatin remodeling (see S2 Table), which belong to the largest groups of sumoylated proteins that are targeted by Siz1 and Siz2. Taken together, the effects of *esc2-D430R* on intracellular sumoylation are similar to those of *mms21* and *esc2Δ*, suggesting that Esc2 recruits Ubc9 to facilitate Mms21 to preferentially target certain nucleolar proteins, the SMC and MCM complexes with Siz1 and Siz2 acting in a parallel pathway.

### Esc2 regulates Mms21-dependent sumoylation independent of Siz1 and Siz2

The fact that the *siz1Δ siz2Δ esc2-D430R* triple mutant is viable presents an opportunity to examine the effect of *esc2-D430R* further. To do so, quantitative SUMO proteomics was performed to compare the levels of sumoylated proteins in the *siz1Δ siz2Δ esc2-D430R* triple mutant and *siz1Δ siz2Δ* double mutant. Because Siz1 and Siz2 are known to regulate the bulk of sumoylated proteins [4, 6], relatively fewer sumoylated proteins were identified and quantified in this experiment (compare S2 and S3 Tables). Nevertheless, the preferred targets of Mms21 were identified and quantified with a comparable number of peptides, allowing us to evaluate the effect of *esc2-D430R*. Several observations are noteworthy. First, although Siz1 and Siz2 are known to play a major role in regulating the sumoylation of Septins and PCNA [6, 31], their sumoylated forms are still identified in the *siz1Δ siz2Δ* mutant, albeit at a lower level, as indicated by the fewer number of peptides identified for these proteins (Fig 6 and compare S2 and S3 Tables). This suggests that Siz1/Siz2 and Mms21 have partially overlapping activity towards most, if not all targets, in the cell. Second, *esc2-D430R* reduces the sumoylation levels of Mms21-preferred targets in the *siz1Δ siz2Δ* double mutant, including those in the nucleolus (Fig 6B), the SMC proteins (Fig 6C) and the MCM proteins (Fig 6D). Overall, these effects are similar to those caused by *esc2-D430R* in the WT strain background (Fig 5), confirming that Esc2-Ubc9 acts in a pathway independent of Siz1 and Siz2.

To directly evaluate whether the Esc2-Ubc9 interaction facilitates Mms21 substrate-targeting in the cell, we directly compared sumoylation levels in the *mms21-CH* and *esc2-D430R* mutants and found that these mutations cause a similar effect, resulting in modest changes of below 2-fold for most targets (Fig 7). Interestingly, sumoylation levels of multiple MCM subunits are consistently higher in the *esc2-D430R* mutant than the *mms21-CH* mutant, although the effects are modest (~2-fold). Moreover, this trend applies to Smc5, Smc6 and a number of nucleolar proteins (Fig 7C and 7D), but not to the others (Fig 7A, 7B and 7F, and S4 Table). Notably, these effects were quantified based on multiple peptides of each protein. These findings suggest that the Esc2-Ubc9 pathway plays a partial role in regulating Mms21-dependent sumoylation.

## Discussion

Accumulating evidence has suggested that *S. cerevisiae* Esc2 and its ortholog *S. pombe* Rad60 recruit Ubc9 to regulate intracellular sumoylation catalyzed by the Mms21/Nse2 SUMO E3 ligases [4, 20, 28, 29]. Complicating this model is the observation that Esc2 also regulates DNA

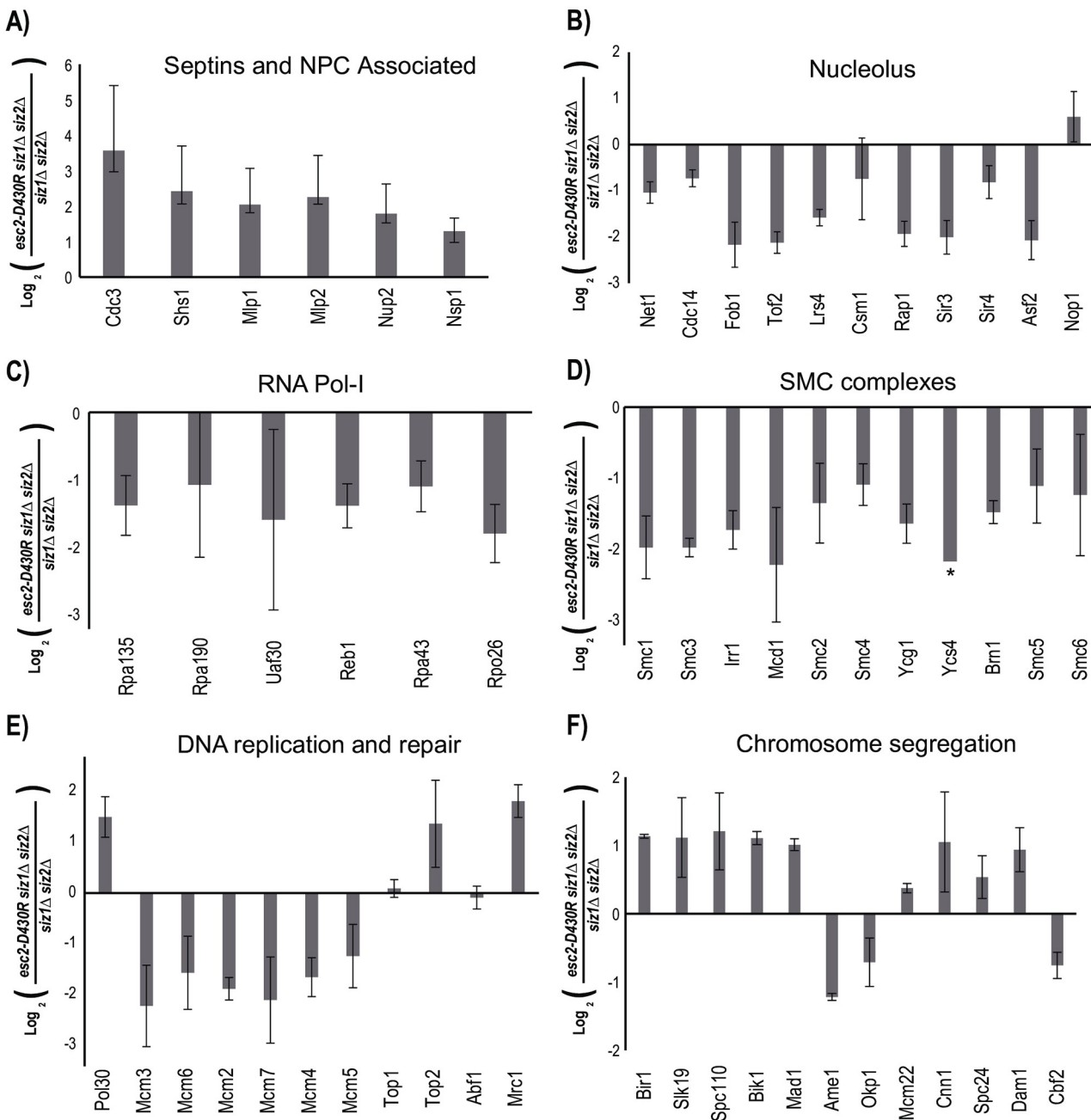

**Fig 6. SUMO proteomic approach to evaluate the effect of the *esc2-D430R* on intracellular sumoylation in cells lacking Siz1 and Siz2, via the comparison of the *siz1Δ siz2Δ esc2-D430R* triple mutant and the *siz1Δ siz2Δ* double mutant.** A) Septins, B) NPC associated, C) Nucleolus and RNA Pol-I, D) the SMC complexes, E) DNA replication and repair and F) Chromosome segregation. Log$_2$ ratios of comparing the relative abundance of each protein in the WT vs the esc2-D430R mutant is shown, along with error bars representing the standard deviation calculated using multiple unique peptides of the same protein (see experimental method for details).

damage repair besides preventing spontaneous genome rearrangements [4, 20–25]. Here, we evaluate the role of the Esc2-Ubc9 interaction, specifically regarding its genetic relationships with the SUMO E3 ligases and its role in regulating intracellular sumoylation in unperturbed cells. Our results demonstrate that together with the Mms21 SUMO E3 ligase, Esc2 recruits

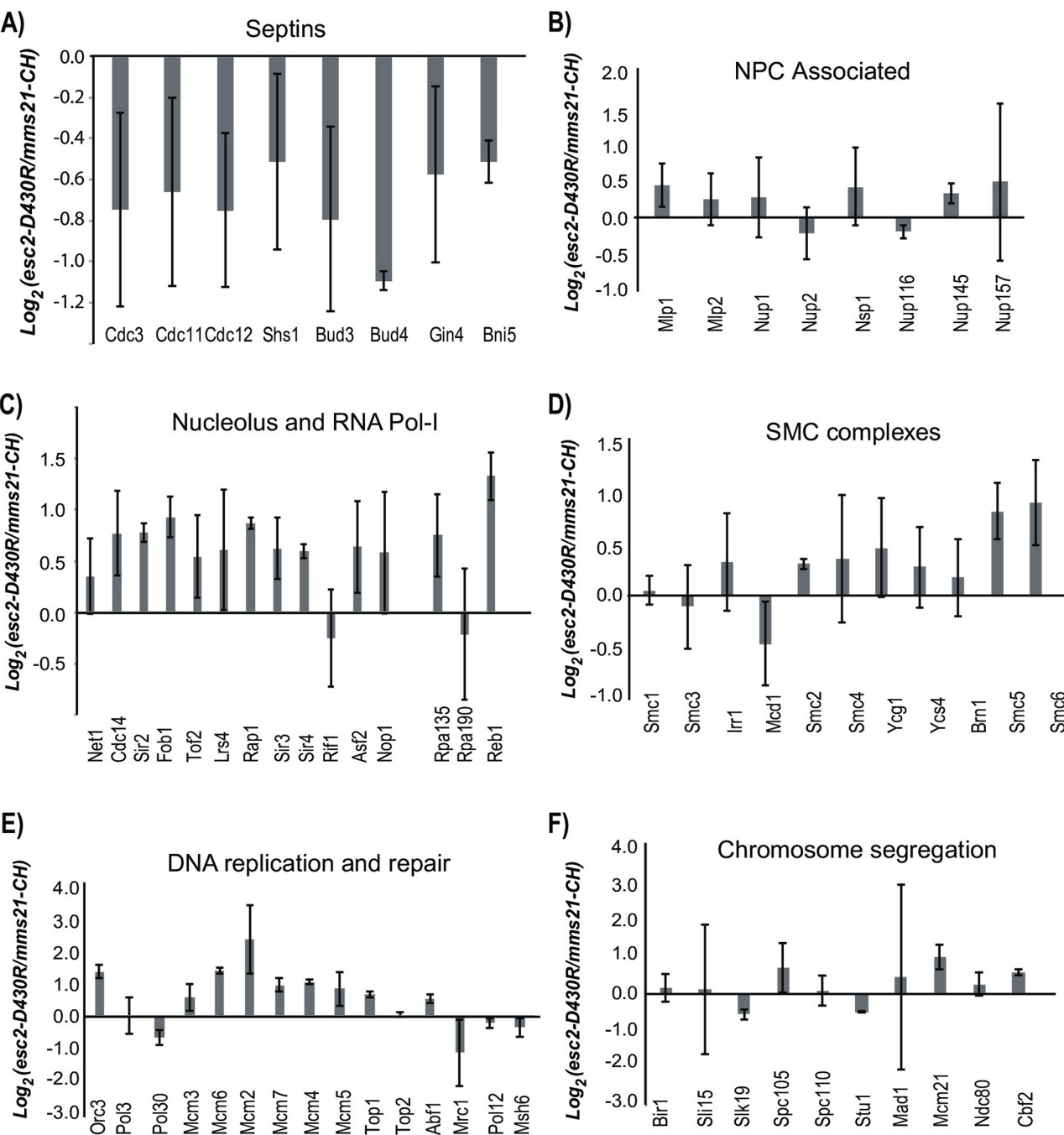

**Fig 7. *esc2-D430R* and *mms21-CH* have similar effects on intracellular sumoylation.** A) Septins, B) NPC associated, C) Nucleolus and RNA Pol-I, D) the SMC complexes, E) DNA replication and repair and F) Chromosome segregation. Log₂ ratios of comparing the relative abundance of each protein in the *esc2-D430R* and *mms21-CH* mutant are shown, along with error bars representing the standard deviation calculated using multiple unique peptides of the same protein (see experimental method for details).

Ubc9 to regulate intracellular sumoylation and suppress dGCRs through a Siz1 and Siz2 independent pathway.

The findings here demonstrate that Esc2 and Mms21 have distinct and shared roles in suppressing dGCRs and responding to genotoxic agents. For example, the *esc2Δ mms21-CH*

double mutant accumulates dGCRs at a higher rate and grows poorly relative to either single mutant (Fig 1). Although the formation of dGCRs in the *esc2Δ* and *mms21-CH* mutants are both mediated by Pol32-, Rad9- and Rad52- dependent recombination processes, *rrm3Δ* results in a further increase of dGCRs in the *esc2Δ* mutant, while it suppresses the dGCRs in the *mms21-CH* mutant [5], indicating that some dGCR suppression attributes of Esc2 are independent of Mms21. In an attempt to separate Esc2's role in the suppression of dGCRs from its role in DNA repair, we analyzed the *esc2-D430R* mutation, which specifically disrupts the Esc2-Ubc9 interaction, similar to the effect of *rad60-E380R* in *S. pombe* [28, 29]. Several lines of evidence support the specific role of the Esc2-Ubc9 interaction in facilitating Mms21. First, the *esc2-D430R* mutant has a synergistic growth defect and rapidly accumulates dGCRs when combined with the *siz1Δ siz2Δ* double mutant, but not with the *mms21-CH* mutant. Second, *esc2-D430R* and *mms21-CH* affects sumoylation in a similar fashion, as Mms21-preferred targets are similarly affected by *esc2-D430R* and *mms21-CH*, while additionally mutating Siz1 and Siz2 does not appreciably alter these *esc2-D430R* specific effects. Currently, it is unclear which features of Esc2 allow it to work together with Mms21, although *S. pombe* Rad60 was previously reported to interact with the Smc5/6 complex where Nse2/Mms21 is a co-subunit [32]. However, the molecular mechanism of this interaction has not been understood. Notably, the conserved di-phenylalanine motif in the unstructured N-terminus of Esc2 appears to regulate its stability and partially contributes to GCR suppression (Fig 2), but we have been unable to detect any interaction between this Esc2 motif and the Smc5/6 complex or others. Alternatively, Esc2 may be recruited to the same chromosomal regions where the Smc5/6-Mms21 complex might be present. For example, Esc2 has been shown to bind to specific DNA structures [24]. In doing so, Esc2 could recruit Ubc9 to those chromosomal regions, allowing Mms21 to act. As such, further studies are needed to distinguish these possibilities or uncover new ones.

Our proteomic analysis here has provided a more complete picture of how Esc2 regulates intracellular sumoylation than our previous study, allowing us to evaluate the specific role of the Esc2-Ubc9 interaction. Esc2's role appears to be independent of Siz1 and Siz2, and it affects a wide range of sumoylated proteins that are preferentially targeted by Mms21. They include certain nucleolar proteins, multiple components of the SMC proteins and the MCM complex (Figs 5–7). In each case, *esc2-D430R* and *mms21-CH* cause highly similar reductions in their sumoylation. Although other possibilities cannot be excluded, these effects are best explained by the idea that Esc2 recruits Ubc9 to specifically facilitate Mms21 activity. Defects in chromosomal DNA replication have been implicated as a major source dGCRs accumulation [33]. It is particularly interesting to observe that the sumoylation of several MCM subunits are similarly down regulated by *esc2-D430R* (Figs 5 and 7), which is similar to our previous observations of the *mms21-CH* mutant [12]. The relatively mild dGCR defect observed in the *esc2-D430R* mutant, compared to *mms21-CH*, suggests that Mms21 has a consistently stronger, albeit modest, role in maintaining MCM sumoylation than Esc2-Ubc9 (Fig 7). Furthermore, *in vivo* residual binding between Esc2-D430R and Ubc9 cannot be ruled out. Regardless, the findings here strongly suggest that the preferred sumoylated targets of Mms21 and Esc2, including MCM and possibly others, play a role in suppressing dGCRs through mechanisms that have yet to be discovered.

## Materials and methods

### Yeast genetic methods

Standard yeast methods were used to construct yeast strains in this study. All integrated mutations were introduced into the chromosomal loci of each gene of interest and confirmed by

DNA sequencing. Double mutants were constructed through genetic crosses and sporulation, yielding multiple independent isolates for genetic analyses. Tetrad dissection was performed on a Singer Instruments MSM 400 and the genotypes of individual spores of each tetrad were confirmed via growth on selective media plates. The strains used are listed in S5 Table, while the plasmids used are listed in S6 Table. Details of plasmid construction are available upon request.

## dGCR assay

GCR analysis was performed as previously described [34]. At least 16 independent isolates of each mutant were examined for the calculation of the median GCR rate. Error bars in the graph represent the upper and lower limits of the 95% confidence intervals of the median. For comparison, the dGCR rates of mutants obtained from previous studies are indicated in the text.

## Biochemical methods

To evaluate Esc2 protein expression levels, 10 mL of log-phase cells ($OD_{600nm} \sim 1.0$) were harvested and whole cell lysate was extracted using glass bead beating. The concentration of protein in cell lysates were normalized using the Bradford assay (Bio-Rad) and TAF-tagged Esc2 proteins were detected on western blot via chemiluminescence method. The antibodies used were a rabbit anti-Protein A primary antibody (1:10,000, Sigma) and a goat anti-Rabbit HRP secondary antibody (1:10,000).

Protein binding assays were performed using C-terminal 6xHIS tagged Ubc9 was expressed in BL21 cells and purified using a Ni-NTA affinity column. N-terminal Protein-A tagged Esc2 WT and Esc2-D430R mutant proteins were similarly expressed and purified with IgG sepharose resin. 40 μg of Esc2 (WT and D430R) was incubated and bound to 20 μL IgG resin; and was subsequently incubated with 100 μg of purified Ubc9 in a final volume of 200 μL for 2 hours on ice. The IgG resins were then washed 5 times with 1 mL PBS (Phosphate-Bu with 0.2% NP-40. After washing, the bound Ubc9 protein was eluted from the IgG resin with 20 μL of 0.1M glycine-HCl and then 20 μl 1% SDS loading buffer for visualization by Coomassie staining.

## Spot assay

Each spot assay was performed as follows: each mutant was grown in 4 ml of YPD liquid media until $OD_{600}$ reached ~2.0. Cell density of the yeast culture was normalized to $OD_{600}$ of 0.2 to ensure equal plating. Four five-fold serial dilutions were then performed for each strain in a sterile 96-well plate using sterile deionized water as diluent. 3 μl of each dilution was spotted onto either YPD or YPD plates containing either hydroxyurea (HU) or methyl methanesulfonate (MMS) of indicated concentrations. All three plates were incubated at 30°C for 2 to 3 days before the representative images were acquired using a Bio-Rad ChemiDoc MP imaging system.

## SUMO proteomics methods

For each stable isotope labeling by amino acids in cell culture (SILAC) experiment, approximately 1L of cells ($OD_{600nm}$ 1.3) from each yeast strain were collected. One strain was labeled using "light" lysine and arginine, while the other was labeled by "heavy" lysine-$C^{13}N^{15}$ and arginine- $C^{13}N^{15}$ (see Supplementary Table legends). Harvested cells were immediately treated with 20 mM iodoacetamide (IAA) in 20 ml PBS buffer for 10 minutes at room temperature to inhibit the SUMO proteases. IAA-treated cells were combined and spun down and decanted to remove excess buffer. To resuspend the combined cell pellet, 20 mL of water, 8 mL of 1 M hydrochloric acid, 4 mL of 10% SDS and 3 mL of 1M sodium phosphate (pH8) were added sequentially; the cell suspension was then transferred to a glass beater and lysed for 10 min at

50% duty cycle at room temperature. The protein extract was transferred into a 50 mL tube and then 8 mL of 1 M NaOH was added to adjust the pH to 8. After addition of 10 mM dithiothreitol (DTT), the sample was heated at 80 ˚C for 10 minutes to reduce proteins. Insoluble material was discarded after centrifugation at 4000 RPM for 10 min and 30 mM IAA was added to the clarified cell lysate to alkylate any free cysteines. Purification of HF-SUMO conjugated proteins was performed by incubating the clarified lysate with 2 mL of Ni-NTA beads (Bio-Rad) for 2 hours at 30 ˚C. The Ni-NTA beads were washed three times with 20 mL of cold PBS-0.2% NP40 (PBSN) and then once with 20 mL of PBSN/0.1% SDS. To elute HF-SUMO conjugated proteins, the Ni-NTA beads were incubated with 4 mL of PBSN + 0.1% SDS and 25 mM EDTA for 10 minutes at room temperature. After collecting the first elution, another 8 ml of PBSN containing 25 mM EDTA and protease inhibitors was added to the Ni-NTA beads to elute any remaining proteins, yielding a total elution volume of ~12 mL. The Ni eluent was then incubated with 200 μL of anti-FLAG M2 beads (Sigma) for 2 hours at 4 ˚C. The anti-FLAG beads were then washed two times with 10 mL of ice-cold PBSN followed by two more times with 1 mL of PBSN. To elute sumoylated proteins 1 μg of recombinant Ulp1 catalytic domain was added to the beads in 1.5 mL PBSN containing 2 mM DTT for 2 hours at 30 ˚C. The Ulp1-eluted sample was then digested by 1ug of trypsin (Promega) at 37 ˚C overnight. Next, the trypsin-digested sample was acidified with 0.2% TFA and then gradually applied to a 100 mg C18 Sep-Pac column (Waters). Peptides were washed three times with 0.3 mL of 0.5% acetic acid and eluted with 0.7 mL 80% acetonitrile/0.5% acetic acid and dried under vacuum centrifugation. The dried peptides were then resuspended in 0.1 mL of 80% acetonitrile/water and fractionated using a hydrophilic interaction liquid chromatography (HILIC) column to generate 10 fractions, as described previously [35]. Each HILIC fraction was dried under vacuum centrifugation, resuspended with 5 μL of 0.5% acetic acid, and injected onto a liquid chromatography tandem mass spectrometry (LC-MS/MS) system for peptide analysis (Thermo: Orbitrap Fusion Lumos Tribrid MS).

Mass spectrometry (MS) data was searched using COMET (Seattle Proteome Center: Trans Proteomic Pipeline) and peptides were quantified using XPRESS (Seattle Proteome Center: Trans Proteomic Pipeline). For database searching, a static modification of 57.021464 Da was added for cysteine residues and differential modifications of 8.014199 Da and 10.00827 Da for lysine and arginine, respectively, were included. During data analysis, low quality peptide identifications were removed by requiring all identified peptides to have a combined total intensity area above $10E^{-3}$ and have a valid XPRESS ratio. Redundant peptide identifications were then removed so that only the most abundant and unique peptides of each protein were kept (see S2–S4 Tables). The relative abundance ratio for each protein was calculated by first summing the light isotope areas and summing the heavy isotope areas of all of the unique peptides for each protein and then generating the relative heavy/light ratio of these summed areas. To obtain a statistical measure, we chose the top 3 unique peptides of each protein that had the highest combined light and heavy areas, and then used these ratios to calculate the standard deviation. In this way, only those proteins identified with 3 or more peptides were included in the calculation. Finally, we queried the results against a list of known SUMO targets to obtain the final results that are shown in S2–S4 Tables.

## Supporting information

**S1 Table. Rates of accumulating dGCRs were measured using fluctuation analysis. At least two independent isolates per mutant were used**. The results are plotted in Figs 1C, 1D, 2F and 7C.
(DOCX)

**S2 Table. MS results of sumoylated proteins found in WT and the *esc2-D430R* mutant.** Unique peptides of each SUMO target are included; their summed intensity was used to calculate its abundance ratio. Light isotope was used to grow the WT cells, while heavy isotope was used to grow the *esc2-D430R* mutant. The top 3 most abundant peptides of each sumoylated protein were used to calculate the standard deviation. (XLSX)

**S3 Table. MS results of sumoylated proteins found in the *siz1Δ siz2Δ* and *siz1Δ siz2Δ esc2-D430R* mutants.** Unique peptides of each SUMO target are included; their summed intensity was used to calculate its abundance ratio. Light isotope was used to grow the *siz1Δ siz2Δ esc2-D430R* mutant, while heavy isotope was used to grow the *siz1Δ siz2Δ* double mutant. The top 3 most abundant peptides of each sumoylated protein were used to calculate the standard deviation. (XLSX)

**S4 Table. MS results of sumoylated proteins found in the *esc2-D430R* and *mms21-CH* mutants.** Unique peptides of each SUMO target are included; their summed intensity was used to calculate its abundance ratio. Light isotope was used to grow the *esc2-D430R* mutant, while heavy isotope was used to grow the *mms21-CH* double mutant. The top 3 most abundant peptides of each sumoylated protein were used to calculate the standard deviation. (XLSX)

**S5 Table. Yeast strains used in this study.** (DOCX)

**S6 Table. Plasmids used in this study.** (DOCX)

**S1 Fig. Original gel and blot images of the results shown in Fig 2C and 2E.** (PDF)

## Acknowledgments

We would like to thank members of the Zhou lab for technical assistance and discussions during the preparation of this work.

## Author Contributions

**Conceptualization:** Raymond T. Suhandynata, Huilin Zhou.

**Data curation:** Raymond T. Suhandynata, Yong-Qi Gao, Ann L. Zhou, Yusheng Yang, Pang-Che Wang, Huilin Zhou.

**Formal analysis:** Raymond T. Suhandynata, Ann L. Zhou, Yusheng Yang, Huilin Zhou.

**Funding acquisition:** Huilin Zhou.

**Investigation:** Raymond T. Suhandynata, Yong-Qi Gao, Ann L. Zhou, Yusheng Yang, Pang-Che Wang, Huilin Zhou.

**Methodology:** Raymond T. Suhandynata, Ann L. Zhou, Huilin Zhou.

**Project administration:** Huilin Zhou.

**Resources:** Huilin Zhou.

**Software:** Ann L. Zhou.

**Supervision:** Huilin Zhou.

**Validation:** Raymond T. Suhandynata.

**Writing – original draft:** Raymond T. Suhandynata, Huilin Zhou.

**Writing – review & editing:** Raymond T. Suhandynata, Huilin Zhou.

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
