## [Decision Letter · Decision Letter 0]

23 Dec 2020

PONE-D-20-36830

Shared and distinct roles of Esc2 and Mms21 in suppressing genome rearrangements and regulating intracellular sumoylation

PLOS ONE

Dear Dr. Zhou,

Thank you for submitting your manuscript to PLOS ONE. After careful consideration, we feel that it has merit but does not fully meet PLOS ONE’s publication criteria as it currently stands. Therefore, we invite you to submit a revised version of the manuscript that addresses the points raised during the review process. 

We look forward to receiving your revised manuscript.

Kind regards,

Anja-Katrin Bielinsky

Academic Editor

PLOS ONE

"This work was supported by the National Institute of General Medical Sciences

437 (NIH R01 GM116897 and NIH S10 OD023498) and University of California

438 CRCC faculty seed grant to HZ, and National Cancer Institute (NCI T32

439 CA009523) Postdoctoral Fellowship to RTS. The funders had no role in study

440 design, data collection and analysis, decision to publish, or preparation of the

441 manuscript."

Reviewers' comments:

Reviewer's Responses to Questions

**Comments to the Author**

1. Is the manuscript technically sound, and do the data support the conclusions?

Reviewer #1: Partly

Reviewer #2: Partly

2. Has the statistical analysis been performed appropriately and rigorously? 

Reviewer #1: No

Reviewer #2: Yes

3. Have the authors made all data underlying the findings in their manuscript fully available?

Reviewer #1: No

Reviewer #2: Yes

4. Is the manuscript presented in an intelligible fashion and written in standard English?

Reviewer #1: Yes

Reviewer #2: Yes

5. Review Comments to the Author

Reviewer #1: The manuscript entitled ”Esc2 prevents genome rearrangements through recruiting the E2 Ubc9 enzyme to regulate sumoylation” by Zhou and colleagues aims to dissect the functional relationship between Esc2 and Mms21. The authors show that esc2∆ and mms21-CH mutations to some extend additively increase HU sensitivity and dGCR rates. The lack of strong synergy between esc2∆ and mms21-CH could indicate that the two proteins act in the same pathway. However most interestingly, the authors find that rrm3∆ has profoundly different effects on the elevated dGCR rates in esc2∆ and mms21-CH mutants, indicating that Esc2 and Mms21 suppress dGCR by fundamentally different mechanisms. The authors conduct SUMO proteomics in a Ubc9-interaction-defective mutant of Esc2 to test the idea that Esc2 recruits Ubc9 to facilitate Mms21-mediated sumoylation. The result is that the esc2-D430R mutation affects global sumoylation patterns in a “similar” manner as mms21-CH but no detailed comparison is provided. The data is of high quality and the research question is timely, but there are too many loose ends to draw mechanistic conclusions and answer the research question. It is therefore my recommendation that the authors provide further data to document the functional relationship between Esc2 and Mms21 or seek a more specialized journal for publication.

Major issues to address:

1. The analysis of the esc2-2FA mutant generates limited excitement with the reviewer since no biological function has been assigned to the residues. The authors elude to the FF residues being involved in a protein-protein interaction. Identification of this interaction would greatly stregthen this data.

2. Expression of ecs2-2FA from a high-copy plasmid rescues plating efficiency of esc2∆ cells and Esc2 protein levels but does not suppress dGCR. Why not?

3. What is the functional role of MCM sumoylation in the context of dGCR?

4. The conclusion that Mms21 and Esc2 work together in targetting proteins for SUMOylation is supported by the result that esc2-D430R affects global sumoylation patterns in a “similar” manner as mms21-CH but no systematic statistical comparison is provided to document the correlation between the two data sets. The conclusion would also benefit from an analysis of the SUMO proteomics of the esc2∆ mms21-CH double mutant?

5. Since both Esc2 and Mms21 interact with Ubc9, it is unclear why Esc2 would be required to recruit Ubc9 to facilitate Mms21-mediated sumoylation.

Reviewer #2: Comments to the authors

In this paper, Suhandynata et al. report data that advance the understanding of the role of Esc2 in genome stability and the SUMO pathway. They examined genetic interactions between esc2 and SUMO mutants in growth and survival, in suppressing duplication-mediated gross chromosomal rearrangements (dGCRs) and in altering the dynamic of the SUMO proteome. They generated new mutants of esc2 that offer insights into the role of Esc2 in genome stability through its functional cooperation with SUMO ligases. The authors conclude that Esc2 interacts with Ubc9 to facilitate Mms21-dependent sumoylation. They also argue that Esc2 and Mms21 have independent functions in regulating genome stability. The data in this manuscript support the authors’ proposed model and are of interest to the field. It is intriguing that Esc2 may play a role in conferring specificity to a SUMO ligase in S. cerevisiae, similar to the model proposed in S. pombe. However, a few major points should be addressed to improve the rigor of the study.

First, it is important to substantiate the authors’ claim that Esc2-Ubc9 interaction mediates Mms21-dependent sumoylation. Based on the data provided, one can only deduce a model in which Esc2 influences sumoylation of Mms21 targets, potentially by interacting with Ubc9. The writing should be carefully revised to accurately reflect the evidence available or additional experiments (suggested below) should be performed to substantiate the claim. Second, the authors should perform a systematic comparison of previous mass spectrometry data from mms21-11 and mms21-CH mutants (or at least from one of these mutants) to the current data from esc2-D430R mutants. Third, the authors should discuss the differences between esc2-D430R and rad60-E380R mutants. Data from both mutants support a model in which Esc2 (or its S.pombe homolog) mediates sumoylation of Mms21-specific targets. However, there are noteworthy differences. Please refer to the detailed suggestions outlined below.

Major issues

1. Through sequence alignment, the authors identified D430 in Esc2 as a candidate amino acid residue that mediates Esc2’s interaction with Ubc9 in S. cerevisiae. The corresponding residue in Rad60 in S. pombe (E380) has been shown to mediate Rad60-Ubc9 interaction both in vitro and in vivo (Prudden et al., 2009).

Fig. 3C in this paper only provides the evidence that mutating D430 to R disrupts the interaction between Esc2 and Ubc9 in vitro using purified proteins expressed in bacteria. Before this manuscript, Sollier et al. reported yeast-two hybrid assay data to demonstrate Ubc9 and Esc2 interaction (Sollier et al., 2009). To my knowledge, Ubc9-Esc2 interaction has not been examined under endogenous conditions. The authors should perform a co-immunoprecipitation of Esc2 (wild-type or D430R mutant) and Ubc9 to confirm that Ubc9 and Esc2 interact and that D430R mutation can indeed disrupt this interaction under physiologically relevant conditions in S. cerevisiae. This is a crucial experiment to substantiate the authors’ argument that “together with the Mms21 SUMO E3 ligase, Esc2 recruits Ubc9 to regulate intracellular sumoylation and suppress dGCRs.”

2. In S. pombe, it has been proposed that Rad60-Ubc9 interaction promotes sumoylation of proteins targeted by Mms21 (Prudden et al., 2011). Several lines of evidence support this model. First, similar to nse2-SA (Mms21 mutant deficient in E3 SUMO ligase activity), rad60-E380R mutants are highly sensitive to several treatments that challenge genome stability (HU, UV, CPT and MMS) (Prudden et al., 2009). Second, Rad60 physically interacts with the Smc5/6 complex in S. pombe (Boddy et al., 2003). Third, rad60-E380R mutants and mutants of Pli1, another E3 SUMO ligase, exhibit synthetic lethality (Prudden et al., 2009). Fourth, rad60 mutants and mutants in the Smc5/6 complex exhibit similar phenotypes (Miyabe et al., 2006).

In this study, the authors proposed a similar model in which Esc2 facilitates Mms21-dependent sumoylation. This model was derived from the following observations: 1) Both mms21-CH and esc2� mutants accumulate X-shaped structures and exhibit dGCR (Sollier et al., 2009; Branzei et al., 2006; Albuquerque et al., 2013 and this paper); 2) esc2-D430R mutants show synergistic defects with siz1� siz2� double mutants and mms21 mutants are synthetically sick with siz1� or siz2� mutants (Zhao and Blobel, 2005 and this paper); 3) mms21-CH and esc2 mutants show epistatic interactions in HU sensitivity and generation of dGCR (this paper); 4) the profile of the SUMO proteome in mms21 and esc2-D430R mutants are similar (Albuquerque et al., 2013; de Albuquerque et al., 2016 and this paper).

While these data substantially support the model, there are some disconcerting issues. For instance, evidence demonstrating that esc2-D430R mutants exhibit similar sensitivity to genome instability agents as mms21 mutants do or discussion regarding this issue is lacking in the current manuscript. Based on the data in this manuscript, esc2-D430R mutants do not exhibit any sensitivity to HU whereas, mms21-CH mutants do (Fig. 3F and 6A). Similarly, esc2� cells are not sensitive to HU (Fig 1A). However, these mutants are sensitive to MMS similar to mms21 mutants (Sollier et al., 2009; Choi et al., 2010 and Urulangodi et al., 2015). Assuming that Mms21 targets are at least partially responsible for resisting genotoxic agents, esc2-D430R mutants should exhibit some sensitivity to some of these agents. In S. pombe, rad60-E380R mutants show similar sensitivity to damaging agents when compared to nse2-SA mutants (Prudden et al., 2009). A possible explanation may lie in the extent to which Mms21 depends on Esc2. The authors did suggest that “Mms21 retains partial activity and partially catalyzes intracellular sumoylation without assistance from Esc2” based on the dGCR data but did not address the discrepancy in sensitivity data.

The authors should discuss these issues as part of their Discussion, contrasting observations for Esc2 and Rad60 as well as highlighting specific loopholes in their proposed model. As part of this discussion, they should cite the Rad60-Ubc9 model that was first proposed in S. pombe. In addition, performing an experiment comparing the sensitivity of mms21-CH and esc2-D430R to MMS treatment (and potentially other genotoxic agents) would be helpful to better elucidate the role of Esc2 in regulating Mms21. Both mms21-CH and esc2� mutants have been shown to be sensitive to MMS (for example, Varejão et al., 2018; Sollier et al., 2009, Choi et al., 2010 and Urulangodi et al., 2015).

3. The authors should provide a systematic comparison of the mass spectrometry data from mms21-11 and mms21-CH mutants (Albuquerque et al., 2013; de Albuquerque et al., 2016) to the esc2-D430R data. Although they described that similar SUMO targets were identified in both mutants, the extent to which how much of this overlap occurs is not entirely apparent to the reader. For example, how many targets are common in both mutants (esc2 vs. mms21 mutants)? Do most of these targets show the same trend? How many of them are unique to each mutant? In addition, it would be helpful to present a side-by-side comparison of the amount of increase or decrease in sumoylation of the common targets. Providing a systematic comparison would minimize misunderstanding. More importantly, this analysis can illustrate to what extent Esc2 is crucial for Mms21-specific SUMO conjugates. Do majority of Mms21 targets need Esc2? Do they get sumoylated to some extent even in the absence of Esc2? Answer to these questions can help in addressing issues raised under #2 (why esc2-D430R mutants do not exhibit any sensitivity to HU like mms21-CH mutants).

On a related note, throughout the paper, the authors should be clear about which data set they referred to when they described the similarity between esc2-D430R dataset and Mms21 SUMO targets. For example, they cited Albuquerque et al., 2013 paper in Line 293-294, however, stated the Mms21 mutant as mms21-CH. Based on the information provided in the 2013 paper, mms21-11 mutant was used for the mass spectrometry study.

Line 293-294: “effects resemble those observed for the esc2� and mms21-CH mutants [4], suggesting that….”

In this sentence, I understand that the authors were referring to the mass spectrometry study described in Albuquerque et al., 2013 (reference number 4 in their reference list). If that is the case, mms21-CH should be replaced with mms21-11.

4. One suggested experiment is to express Mms21-Ubc9 fusion protein in esc2-D430R genetic background. This mutant is described in Bermúdez-López et al., 2015. Introduction of this fusion protein artificially recruits Ubc9 to facilitate Mms21 sumoylation of its targets, bypassing the need for a mediator (Bermúdez-López et al., 2015). This experiment can provide better insights into the idea that Esc2-Ubc9 interaction is responsible for sumoylation of Mms21 targets.

Minor issues

Line 111-113: “….there is currently no direct evidence demonstrating that MCM sumoylation plays a role in suppressing dGCRs, as the involvement of other Mms21-preferred targets cannot be excluded.”

The authors set up the question as if the paper was going to demonstrate if MCM sumoylation is responsible for suppressing dGCRs. While this is an intriguing point, it is not a question that the data in this paper are able to address. It is relevant to raise this point in Discussion, but the authors might consider reframing this point in their Introduction.

Line 151-152: “DNA lesions caused DNA damage agents”: Did the authors mean to state “DNA lesions caused by DNA damaging agents”?

Line 207: “to test it role”: It should read “to test its role.”

Line 383-384: This sentence appears to be a part of the previous one.

Line 396: esc2-D430A was mentioned. It should be esc2-D430R.

References

Albuquerque CP, Wang G, Lee NS, Kolodner RD, Putnam CD, Zhou H. Distinct SUMO ligases cooperate with Esc2 and Slx5 to suppress duplication-mediated genome rearrangements. PLoS Genet. 2013;9(8):e1003670. doi: 10.1371/journal.pgen.1003670. Epub 2013 Aug 1. Erratum in: PLoS Genet. 2016 Aug;12(8):e1006302. PMID: 23935535; PMCID: PMC3731205.77; PMCID: PMC2893993.

Bermúdez-López M, Pociño-Merino I, Sánchez H, Bueno A, Guasch C, Almedawar S, Bru-Virgili S, Garí E, Wyman C, Reverter D, Colomina N, Torres-Rosell J. ATPase-dependent control of the Mms21 SUMO ligase during DNA repair. PLoS Biol. 2015 Mar 12;13(3):e1002089. doi: 10.1371/journal.pbio.1002089. PMID: 25764370; PMCID: PMC4357442.

Boddy MN, Shanahan P, McDonald WH, Lopez-Girona A, Noguchi E, Yates III JR, Russell P. Replication checkpoint kinase Cds1 regulates recombinational repair protein Rad60. Mol Cell Biol. 2003 Aug;23(16):5939-46. doi: 10.1128/mcb.23.16.5939-5946.2003. PMID: 12897162; PMCID: PMC166335.

Branzei D, Sollier J, Liberi G, Zhao X, Maeda D, Seki M, Enomoto T, Ohta K, Foiani M. Ubc9- and mms21-mediated sumoylation counteracts recombinogenic events at damaged replication forks. Cell. 2006 Nov 3;127(3):509-22. doi: 10.1016/j.cell.2006.08.050. PMID: 17081974.

Choi K, Szakal B, Chen YH, Branzei D, Zhao X. The Smc5/6 complex and Esc2 influence multiple replication-associated recombination processes in Saccharomyces cerevisiae. Mol Biol Cell. 2010 Jul 1;21(13):2306-14. doi: 10.1091/mbc.e10-01-0050. Epub 2010 May 5. PMID: 204449.

de Albuquerque CP, Liang J, Gaut NJ, Zhou H. Molecular Circuitry of the SUMO (Small Ubiquitin-like Modifier) Pathway in Controlling Sumoylation Homeostasis and Suppressing Genome Rearrangements. J Biol Chem. 2016 Apr 15;291(16):8825-35. doi: 10.1074/jbc.M116.716399. Epub 2016 Feb 26. PMID: 26921322; PMCID: PMC4861450.

Miyabe I, Morishita T, Hishida T, Yonei S, Shinagawa H. Rhp51-dependent recombination intermediates that do not generate checkpoint signal are accumulated in Schizosaccharomyces pombe rad60 and smc5/6 mutants after release from replication arrest. Mol Cell Biol. 2006 Jan;26(1):343-53. doi: 10.1128/MCB.26.1.343-353.2006. PMID: 16354704; PMCID: PMC1317627.

Prudden J, Perry JJ, Arvai AS, Tainer JA, Boddy MN. Molecular mimicry of SUMO promotes DNA repair. Nat Struct Mol Biol. 2009 May;16(5):509-16. doi: 10.1038/nsmb.1582. Epub 2009 Apr 12. PMID: 19363481; PMCID: PMC2711901.

Prudden J, Perry JJ, Nie M, Vashisht AA, Arvai AS, Hitomi C, Guenther G, Wohlschlegel JA, Tainer JA, Boddy MN. DNA repair and global sumoylation are regulated by distinct Ubc9 noncovalent complexes. Mol Cell Biol. 2011 Jun;31(11):2299-310. doi: 10.1128/MCB.05188-11. Epub 2011 Mar 28. PMID: 21444718; PMCID: PMC3133251.

Sollier J, Driscoll R, Castellucci F, Foiani M, Jackson SP, Branzei D. The Saccharomyces cerevisiae Esc2 and Smc5-6 proteins promote sister chromatid junction-mediated intra-S repair. Mol Biol Cell. 2009 Mar;20(6):1671-82. doi: 10.1091/mbc.e08-08-0875. Epub 2009 Jan 21. PMID: 19158389; PMCID: PMC2655255.

Varejão N, Ibars E, Lascorz J, Colomina N, Torres-Rosell J, Reverter D. DNA activates the Nse2/Mms21 SUMO E3 ligase in the Smc5/6 complex. EMBO J. 2018 Jun 15;37(12):e98306. doi: 10.15252/embj.201798306. Epub 2018 May 16. PMID: 29769404; PMCID: PMC6003635.

Zhao X, Blobel G. A SUMO ligase is part of a nuclear multiprotein complex that affects DNA repair and chromosomal organization. Proc Natl Acad Sci U S A. 2005 Mar 29;102(13):4777-82. doi: 10.1073/pnas.0500537102. Epub 2005 Feb 28. Erratum in: Proc Natl Acad Sci U S A. 2005 Jun 21;102(25):9086. PMID: 15738391; PMCID: PMC555716.

6. PLOS authors have the option to publish the peer review history of their article (what does this mean?). If published, this will include your full peer review and any attached files.

Reviewer #1: No

Reviewer #2: No

---

## [Author Response · Author response to Decision Letter 0]

21 Jan 2021

Overall Responses to the Reviewers:

First, we would like to thank both reviewers for their helpful and constructive comments. We have made substantial revisions to the manuscript to address them. Major changes include a new SUMO proteomics experiment to directly compare sumoylation levels in the esc2-D430R and mms21-CH mutants (see New Figure 7). This new result shows that these mutants have similar effects on intracellular sumoylation, supporting the model that the Esc2-Ubc9 pathway functions in the same pathway as Mms21. Additionally, we renamed the old Figure 6 to the new Figure 4, which now includes a new result about the DNA damage sensitivity of various esc2 and mms21 mutants as requested by the reviewers. Specific responses to each reviewer’s comments are discussed below.

Responses to Reviewer 1 comments:

1&2) The conserved FF motif of Esc2 is important for its protein stability in the cell, and the GCR phenotype of esc2-2FA points to other functions that cannot be explained by protein abundance alone. We do not know whether this FF motif of Esc2 mediates protein-protein interactions, since both our candidate testing and unbiased pull-down MS experiments have not revealed any promising candidates. We also cannot exclude the possibility that this FF motif of Esc2 may play other roles that do not involve protein-protein interaction. However, two reasons promoted us to report this finding. First, we feel it is always important to report novel findings that can stimulate further studies by the research community. Second, this result does not adversely affect the main conclusion of this study. 

3) To date, the function of MCM sumoylation in preventing GCRs is unknown. The findings that Mms21 and Esc2 regulate MCM sumoylation suggest that future study to investigate this topic is potentially interesting, but it is beyond the scope of this study. Because of this, we have removed the discussion about this in the introduction section to avoid a misleading impression about the focus of our study.

4) See new Figure 7 for a direct comparison of sumoylation levels in the esc2-D430R and mms21-CH mutants. This new result shows that the levels of sumoylated proteins are similar with relatively modest differences (<2 fold) between these mutants. Because of this, a statistical analysis of our prior published MS findings is no longer needed. 

5) We do not know why Esc2 has evolved to facilitate Mms21’s function by recruiting Ubc9. It is important to emphasize here that Esc2 partially contributes to Mms21’s function. This is supported by the overall milder defects of esc2-D430R than mms21-CH in cell growth, and DNA damage sensitivity and accumulation of GCRs, which correlate with the modestly higher sumoylation levels of Mms21-preferred targets such as MCM in the esc2-D430R mutant than in mms21-CH mutant. However, the functional relevance of MCM sumoylation is a subject of future study.

Responses to Reviewer 2 comments:

1) Compelling evidence describing the Esc2(Rad60)-Ubc9 binding has been reported from the structural study in S. pombe. Here, we confirmed that this interaction is conserved in S. cerevisiae. We have attempted co-IP experiments to detect the binding between endogenously expressed Esc2 and Ubc9, but have so far been unable to detect this interaction. This unpublished observation, combined with the two-hybrid over-expression study by Sollier et al, and the interaction between S. pombe Rad60 and Ubc9, which was also detected using over-expressed proteins, collectively suggest that this interaction may have a relatively modest affinity. As is often the case, the ability to detect protein-protein interactions of modest affinity via a co-IP experiment depends on the levels of proteins in the cell lysate and the non-equilibrium washing conditions used in such experiments. It is important to emphasize here that a modest affinity does not exclude the possibility that they do interact in the cell, which is “molecularly crowded”. Moreover, genetic and proteomic evidence presented in this study and the related studies of Rad60 in S. pombe validate the existence of Esc2-Ubc9 interaction. The observation that a single point mutation of a conserved residue in Esc2/Rad60 affects Ubc9 binding and causes a variety of in vivo phenotypes demonstrates the specificity of this interaction and its functions in vivo. 

2) We appreciate the reviewer’s effort in detailing the similarity of the findings in S. pombe and our findings here in S. cerevisiae. A more explicit discussion of these results and the proposed model are included in the introduction section. Despite some differences in the severity of phenotypes that could be due to the different model organisms used, the overall mechanism is conserved. We have also included the MMS sensitivity of various esc2 and mms21 mutants (new Figure 4B), showing that esc2-D430R, unlike esc2Δ or mms21-CH, is not sensitive to MMS and does not further elevate the sensitivity of the mms21-CH mutant. Combined with the GCR phenotype and genetic interaction with siz1Δ siz2Δ mutant, esc2-D430R has a less severe phenotype than mms21-CH. 

3) We have included a new MS result to compare sumoylation levels in the esc2-D430R and mms21-CH mutants (new Figure 7). This new result provides further support for their similar effects on essentially all sumoylated proteins detected in these mutants. The modestly stronger reduction of sumoylation levels of MCM by mms21-CH than esc2-D430R may contribute to its observed stronger phenotypes in genome maintenance, but the investigation into the function of MCM sumoylation is beyond the scope of this study.

4) The use of a Mms21-Ubc9 fusion protein would cause an unintended caveat, as Mms21 can no longer dissociate from Ubc9, which obviously occurs in the WT cell. While this maneuver could artificially enhance sumoylation of certain Mms21-targets in the esc2-D430R mutant, we feel that the new result in Figure 7 has already provided compelling evidence for the role of Esc2-Ubc9 in regulating Mms21 substrate targeting in the cell.

Minor points:

1) Line 111-113: We agree that the study of MCM sumoylation in suppressing GCR is outside the scope of this study. Thus, this discussion has been removed from the introduction and is only briefly commented on at the end of the discussion section.

2) Line 151-152, corrected.

3) Line 207, corrected. 

4) Line 383-384, corrected.

5) Line 396, corrected.

---

## [Editor Report · Decision Letter 1]

2 Feb 2021

Shared and distinct roles of Esc2 and Mms21 in suppressing genome rearrangements and regulating intracellular sumoylation

PONE-D-20-36830R1

Dear Dr. Zhou,

We’re pleased to inform you that your manuscript has been judged scientifically suitable for publication and will be formally accepted for publication once it meets all outstanding technical requirements.

Kind regards,

Anja-Katrin Bielinsky

Academic Editor

PLOS ONE
---

## [Editor Report · Acceptance letter]

4 Feb 2021

PONE-D-20-36830R1 

Shared and distinct roles of Esc2 and Mms21 in suppressing genome rearrangements and regulating intracellular sumoylation 

Dear Dr. Zhou:

I'm pleased to inform you that your manuscript has been deemed suitable for publication in PLOS ONE. Congratulations! Your manuscript is now with our production department. 

Kind regards, 

on behalf of

Dr. Anja-Katrin Bielinsky 

Academic Editor

PLOS ONE